# A Comprehensive Prognostic Analysis of Tumor-Related Blood Group Antigens in Pan-Cancers Suggests That *SEMA7A* as a Novel Biomarker in Kidney Renal Clear Cell Carcinoma

**DOI:** 10.3390/ijms23158799

**Published:** 2022-08-08

**Authors:** Yange Wang, Chenyang Li, Xinlei Qi, Yafei Yao, Lu Zhang, Guosen Zhang, Longxiang Xie, Qiang Wang, Wan Zhu, Xiangqian Guo

**Affiliations:** 1Department of Preventive Medicine, Institute of Biomedical Informatics, Bioinformatics Center, Henan Provincial Engineering Center for Tumor Molecular Medicine, School of Basic Medical Sciences, Academy for Advanced Interdisciplinary Studies, Henan University, Kaifeng 475004, China; 2Department of Anesthesia, Stanford University, Stanford, CA 94305, USA

**Keywords:** tumor-associated antigens (TAAs), blood group antigen, cancer biomarker, prognosis, pan-cancers

## Abstract

Blood group antigen is a class of heritable antigenic substances present on the erythrocyte membrane. However, the role of blood group antigens in cancer prognosis is still largely unclear. In this study, we investigated the expression of 33 blood group antigen genes and their association with the prognosis of 30 types of cancers in 31,870 tumor tissue samples. Our results revealed that blood group antigens are abnormally expressed in a variety of cancers. The high expression of these antigen genes was mainly related to the activation of the epithelial-mesenchymal transition (EMT) pathway. High expression of seven antigen genes, i.e., *FUT7*, *AQP1*, *P1*, *C4A*, *AQP3*, *KEL* and *DARC*, were significantly associated with good OS (Overall Survival) in six types of cancers, while ten genes, i.e., *AQP1*, *P1*, *C4A*, *AQP3*, *BSG*, *CD44*, *CD151*, *LU*, *FUT2*, and *SEMA7A*, were associated with poor OS in three types of cancers. Kidney renal clear cell carcinoma (KIRC) is associated with the largest number (14 genes) of prognostic antigen genes, i.e., *CD44*, *CD151*, *SEMA7A*, *FUT7*, *CR1*, *AQP1*, *GYPA*, *FUT3*, *FUT6*, *FUT1*, *SLC14A1*, *ERMAP*, *C4A*, and *B3GALT3*. High expression of *SEMA7A* gene was significantly correlated with a poor prognosis of KIRC in this analysis but has not been reported previously. *SEMA7A* might be a putative biomarker for poor prognosis in KIRC. In conclusion, our analysis indicates that blood group antigens may play functional important roles in tumorigenesis, progression, and especially prognosis. These results provide data to support prognostic marker development and future clinical management.

## 1. Introduction

Human blood groups can be classified based on the presence or absence of certain heritable antigenic substances (mainly glycoproteins) on the erythrocyte membrane [1,2]. The ABO blood group system and RH blood group system are the most widely used in clinical practice. Many studies found the ABO or RH blood group to be associated with the risk of diseases [3]. For example, the O and RH-blood groups are reported to be associated with a slightly lower risk for SARS-CoV-2 infection and severe COVID-19 illness [4], and mortality risk in A-blood group individuals was significantly higher than in the O-blood group individuals [5]. Blood type ABO was also reported to be associated with risk of cancers. Compared to blood type O, blood type B and AB was associated with an increased risk of esophageal squamous cell carcinoma. Blood type A and AB were significantly associated with gastric noncardia adenocarcinoma [6]. Blood type A was associated with an increased risk of gastric cancer [7]. These results imply that blood groups are important for clinical management and consultation.

In addition to the ABO and RH blood group systems, more than 20 other blood group systems were reported, and some blood group antigens have been found to be significantly overexpressed in cancer cells when compared to normal cells, playing an important role in the development of tumors as tumor-associated antigens (TAAs) [8]. For example, *BSG* (*CD147*), an OK blood group antigen system gene, is described to be upregulated in several human cancers [9,10], including cervical squamous cell carcinoma [11], increased *BSG* expression is significantly associated with a higher pathologic stage, tumor size, invasion depth, lymphoid infiltration, and a number of other prognostic parameters in cervical squamous cell carcinoma and renal clear cell carcinoma [12,13,14]. *AQP1*, a Colton blood group antigen system gene, is over-expressed in a variety of human cancers, including brain, breast, lung and nasopharyngeal cancers [15,16,17,18,19]. Additionally, the expression of *AQP1* has been reported to be associated with poor prognosis and other clinical characteristics such as histological grade and status of lympho-vascular invasion and nodal involvement in cervical carcinoma and colon cancer [20,21,22], etc.

Although there have been sporadic studies of these blood group genes in tumor prognosis, the systemic roles of these blood group antigens in the prognosis of pan-cancers remains to be established. In this study, we collected 33 antigenic genes from 29 blood group systems, and performed a comprehensive analysis of the expression of 33 blood group antigenic genes and their prognostic values in pan-cancers. Finally, we suggested that the gene *SEMA7A* as a putative new biomarker for poor prognosis in KIRC. This study provides data support for blood group systems in cancer prognosis.

## 2. Results

### 2.1. Blood Group Antigen Collection

A comprehensive review of the available studies is presented in the flow diagram in Figure 1A. A total of 33 blood group antigen genes which belong to 29 blood group systems were included for a cancer prognosis in this study (Figure 1B) derived from 39 studies. These studies showed that 15 blood group antigen genes including *CD151*, *AQP1*, *CD55*, *AQP3*, *GYPC*, *FUT6*, *FUT1*, *ABO*, *GCNT2*, *FUT7*, *FUT3*, *FUT2*, *DARC*, *CR1* and *BSG* were reported as prognostic biomarkers in cancers. For example, *CD151*, a Raph blood group antigen system gene which has been studied most frequently, was reported to be associated with poor prognosis in breast cancer and glioblastoma [23,24]. *AQP1*, a Colton blood group antigen system gene, was reported to be associated with poor prognosis in colon cancer and lung adenocarcinoma [25,26], while seven of the fifteen genes were reported to be prognostic in one tumor type (*GCNT2*, *FUT7*, *FUT3*, *FUT2*, *DARC*, *CR1* and *BSG*, Figure 1C). However, the prognostic value of the other 18 blood group antigens in cancers is unclear.

### 2.2. The Abnormal Expression of Blood Group Antigen Genes in Pan-Cancers

To explore the expression variations of different blood group antigen genes in pan-cancers, the Oncomine database was analyzed. As shown in Figure 2, all blood group antigen genes were differentially expressed in 20 cancer types, among which the number of differentially expressed blood group genes was lowest in myeloma, with only eight genes in total, while the highest number was twenty-eight genes in kidney cancer. Among all the antigen genes, the expression levels of *CD44* increased more in tumors compared to normal tissues, while *AQP3* decreased the most in tumors (top 1). The expression patterns of the blood group antigens in the Oncomine datasets are shown in the heatmap (Figure 2). These results suggest that blood group antigen genes may play important roles in the development of tumors.

### 2.3. Prognostic Role of Blood Group Antigen Genes in Pan-Cancers

Blood group antigen genes have been reported to be involved in tumor development, progression and prognosis [12,13,14,20,21,22]. To further explore the prognostic significance of all the blood group genes in pan-cancers, we collected 30,052 clinical cancer cases with both transcriptomic profiles and long-term follow-up information using LOGpc webtool, an online tool we built to analyze outcome and prognostic biomarker development. In addition, we obtained the data of 1818 cases of KICH, PCPG, PRAD, READ, MESO, TGCT and THCA (absent in LOGpc database) with the TISIDB webtool, to evaluate the relationship between blood group antigen genes and overall survival (OS) in pan-cancers. The results showed that the antigen genes were significantly associated with different outcomes in the analyzed cancers. Briefly, the high expression of seven blood group antigen genes, i.e., *FUT7*, *AQP1*, *P1*, *C4A*, *AQP3*, *KEL*, and *DARC*, was significantly associated with good prognosis in six types of cancers, i.e., KIRC, HNSC, LUAD, CESC, SARC, MESO. In addition, 10 genes, i.e., *AQP1*, *P1*, *C4A*, *AQP3*, *BSG*, *CD44*, *CD151*, *LU*, *FUT2*, and *SEMA7A*, were associated with the poor prognosis in three types of cancers, i.e., KIRC, LGG and UVM, in the TCGA dataset (Figure 3A). *AQP1*, a gene of Colton blood system, has prognostic significance in most cancer types, including the association of good outcomes in seven types of cancers, i.e., HNSC, SARC, LIHC, KIRP, KICH, MESO, KIRC and poor outcomes in three cancers (LUSC, LGG, UVM). Some of the results are presented in Figure 3B–E. However, the blood group gene, *GYPE*, which belongs to the MNS antigen system, has no significance of prognostication in all cancer types (Figure 3F). There is no indication of OS significance in CHOL, PCPG or PRAD cancers for all blood group antigen genes. Strikingly, approximately 58% (19 of 33) of the analyzed blood group genes have prognostic value in KIRC (Figure 3G). These results were also validated by multiple independent studies using prognostic study methods (DFI, PFI, DSS and DFS, etc.) (Appendix A), and the comprehensive prognostic outcomes were summarized, including the results for the most often studied gene, *CD151*, which was shown to be associated with a poor prognosis in multiple datasets of LUCA (Appendix A).

The staging and grading systems are important prognostic factors in clinics. To measure the association between blood group genes and stages/grades, and further determine the independent prognostic roles of blood group genes, we analyzed the relationship between the expression of blood group system genes and cancer stage or grade. The results showed that the high expression of six antigen genes, i.e., *FUT1*, *AQP1*, *C4A*, *ART4*, *RHCE* and *DARC*, was significantly associated with the earlier stage of eight types of cancers, i.e., KIRC, LIHC, LUAD, HNSC, PAAD, TGCT, KIRP and LUSC, and associated with the advanced stage of eight types cancers, i.e., BLCA, OV, BRCA, COAD, READ, SKCM, STAD and UCEC. We also noticed that the stages of CHOL, PCPG and PRAD cancers had no significant correlation with the expression of any blood group antigen genes mentioned in this study (Figure 3H). The analysis between blood group gene expression and tumor grade showed that the expression of these antigen genes had significant associations with grades in 10 tumor types, which are LGG, HNSC, STAD, CESC, PAAD, LUAD, OV, UCEC, KIRC and LIHC, and that the high expression of five genes (*SLC14A1*, *P1*, *ERMAP*, *AQP1*, *FUT6*) was associated with lower grades in UCEC and KIRC. On the contrary, high expression of four genes, i.e., *KEL*, *GCNT2*, *P1* and *ERMAP*, was associated with a higher cancer grade in LGG and HNSC (Figure 3I).

### 2.4. SEMA7A Is a Putative New Prognostic Marker for KIRC

The above results indicate that blood group antigen genes have prognostic values in most cancers and can be labeled as prognostic biomarker candidates for further investigation.

Analyzing the relationship between gene expression and cancer-related pathways is valuable in order to understand the mechanism of prognostic molecules. The results of the cancer-related pathway analysis showed that the high expression of these antigen genes was mainly related to the activation of the epithelial–mesenchymal transition (EMT) pathway, whereby *CR1* and *SEMA7A* were involved in up to 41% of cancer types (Appendix A). The high expression of these two genes was shown to have a significant association with OS, stage and grade in KIRC (Appendix A). Since no previous reports suggested *SEMA7A* as a new prognostic biomarker, we chose *SEMA7A* genes for further prognostic analysis in KIRC.

To investigate the role of *SEMA7A* in the development of KIRC, we performed some pilot studies to explore the expression patterns of *SEMA7A* in KIRC tissues and normal kidney tissues, and found that the protein expression of SEMA7A was significantly elevated in KIRC tissues compared to normal tissues in the UALCAN database (Figure 4A); however, the mRNA expression level of *SEMA7A* increased but not significantly in tumors in the TCGA dataset.

Next, we analyzed the relationship between *SEMA7A* expression and clinicopathological parameters in KIRC. As shown in Table 1, high *SEMA7A* expression was closely correlated with advanced tumor stages (III/IV) and grades (G3/G4) (*p* < 0.0001, respectively), but not significantly correlated with age, gender or smoking history. Furthermore, Kaplan–Meier survival curves (Figure 4B–E) revealed that patients with high levels of *SEMA7A* expression exhibited shorter overall survival than those with a lower *SEMA7A* level. Multivariate Cox regression analysis further demonstrated that a more advanced stage and grade of tumor and higher *SEMA7A* mRNA expression were independent unfavorable prognostic factors of OS (*p* < 0.0001) (Table 2). Taken together, these results strongly indicated that *SEMA7A* could be an independent unfavorable prognostic biomarker for KIRC.

To further investigate the role of *SEMA7A* in tumors, the KIRC patients were divided into two subgroups according to the median expression level of *SEMA7A*, and functional enrichment of differential expression genes (DEGs) between the two subgroups was performed. The analysis results showed that the DEGs were mainly enriched in cellular components (CC) of the extracellular matrix, in the biological processes (BP) of regulated cellular activation, and in molecular functions (MF) of receptor regulator activity (Figure 5A). The pathway analysis results showed that the DEGs were mainly enriched in cytokine–cytokine receptor interactions, and the NF-kappa B signaling pathway (Figure 5B).

## 3. Discussion

TAAs (Tumor-associated antigens) play important roles in the development of cancers. Previous studies found that some of the blood group antigens were involved in tumorigenesis and the progression of multiple cancers such as uterine cervical carcinoma, gastric cancer, colorectal cancer, cervical squamous cell carcinoma, clear cell renal cell carcinoma, breast cancer, glioblastomas, lung cancer and nasopharyngeal cancer [12,13,17,18,19,21,23,24,27,28], but the literature lacked a comprehensive study of whether blood group antigens are associated with cancer prognosis.

According to the OS analysis of blood group antigen genes in pan-cancers, most antigen genes have prognostic values in most cancer types. Certain antigen genes have consistent prognostic roles in multiple cancers, such as those of the *FUT* family, including *FUT7*, *FUT3* and *FUT6* from the Lewis antigen system; among them, the prognostic role of *FUT7* in BLCA [29], *FUT3* in clear cell renal cell carcinoma [30], and *FUT* family in acute myeloid leukemia [31] have been validated. Some antigen genes have synergic prognostic roles, such as *FUT2* (ABO blood group system), *ART4* (Dombrock antigen system), and *DARC* (Duffy antigen system) in STAD; among them, the prognostic role of *FUT2* in STAD has been validated [5,6]. Four genes, i.e., *RHD*, *RHCE*, *SLC4A1* and *ACHE* have prognostic value in only one type of cancer, such as *RHD* (Rhesus blood group system) in OV, *RHCE* (Rhesus blood group system) in SARC, (Figure 3A), etc. However, none of the 33 blood group antigen genes have prognostic values in prostate adenocarcinoma (PRAD), cholangiocarcinoma (CHOL), pheochromocytoma and paraganglioma (PCPG); in addition, *GYPE*, from the MNS antigen system has no prognostic significance in all cancers (Figure 3F,G).

It should be noted that KIRC has the largest number of prognostic blood group antigen genes, which include 14 antigen genes, including *CD44*, *CD15*, *SEMA7A*, *FUT7*, *CR1*, *AQP1*, *GYPA*, *FUT3*, *FUT6*, *FUT1*, *SLC14A1*, *ERMAP*, *C4A* and *B3GALT3* (Appendix A). Among them, the high expression of *SEMA7A* was associated with the EMT activation pathway in multiple types of tumors (Appendix A). SEMA7A is the only GPI-anchored semaphorin [32] which plays a role in varied inflammatory diseases such as skin inflammation [33], corneal inflammation [34], multiple sclerosis [35], and liver fibrosis [36]. In addition, *SEMA7A* regulates the proliferation, migration, invasion, lymph formation and angiogenesis of multiple types of tumor cells [37]. A recent study elucidated a novel role of *SEMA7A* in its regulation of EMT in a mammary epithelial cell line. The inhibition of *SEMA7A* in EpRas cells resulted in an impaired ability to undergo TGF-β-induced EMT [38]. In this study, based on the comprehensive prognostic analysis of the blood group antigen genes in pan-cancers, we found that the high expression of *SEMA7A* in KIRC was associated with poor prognosis of KIRC patients, implying that *SEMA7A* may be a novel unfavorable prognostic biomarker for KIRC.

In addition to the prognostic investigation, we also analyzed the mRNA expression variations of these blood group genes in pan-cancers. The results showed that most genes were abnormally expressed in most of the tumors when compared to normal tissues (Figure 4), such as *CD44* expression which is higher in most cancers than normal tissues, indicating that *CD44* may play an important role in the development of tumors. As reported, *CD44* was involved in the metastasis of malignant tumors in several studies [39,40,41].

In addition to the transcriptional expression level of blood type genes, it is also necessary to investigate their protein expression, or various glycosyltransferase enzymes that help attach the sugar molecules to the protein antigens defining the blood type [42]. Therefore, glycosyltransferases expression variation or the changes in glycan structure are also recommended to be investigated in the future.

Overall, our analysis indicated that blood group antigens may play functionally important roles in tumorigenesis, progression, and prognosis. These results provide data support for prognostic biomarker development and future clinical management.

## 4. Materials and Methods

### 4.1. Search Process

To collect the antigen genes of human blood group systems, we performed the search with terms “blood group antigen or system” in PubMed and Google Scholar. As a result, 29 blood type systems with 33 genes were collected. Next, the search strategy combining the terms “(gene symbol) and (cancer OR carcinoma OR tumor) and (prognos* OR surviv*) and (immunohisto* OR protein OR biomarker)” were entered into PubMed (1970–2021) to find relevant reports of the prognostic value of these 33 genes in tumors. As a result, 39 studies with verified prognostic biomarkers were found.

### 4.2. Relationships between Antigen Gene and Cancer Prognosis

The LOGpc system (http://bioinfo.henu.edu.cn/DatabaseList.jsp, accessed on 31 July 2021) including 30,052 tumor cases with both gene expression profiling data and clinical long term follow-up data was used to analyze the relationship between the blood group antigen genes and prognosis in pan-cancers. Prognostic data for the kidney chromophobe (KICH), pheochromocytoma and paraganglioma (PCPG), prostate adenocarcinoma (PRAD), rectum adenocarcinoma (READ), mesothelioma (MESO), testicular germ cell tumors (TGCT) and thyroid carcinoma (THCA) were obtained by employing the TISIDB (http://cis.hku.hk/TISIDB/, accessed on 31 August 2021) in TCGA with 1818 tumor cases. TBtools software was used to draw heatmap for above data [43].

### 4.3. Analysis of Antigen Gene Expression

The Oncomine (https://www.oncomine.org/resource/main.html, accessed on 31 August 2021) webtool was used to analyze the differential gene expression of blood group antigens between tumor and normal tissue regions in pan-cancers. The analysis parameters in Oncomine include *p* value < 0.05, |log2(fold change)| ≥ 1, and gene ranking in the top 10%. In addition, the GEPIA (http://gepia.cancer-pku.cn, accessed on 31 August 2021) webtool was used to analyze the association between gene expression and stages across human cancers. The TISIDB (http://cis.hku.hk/TISIDB/, accessed on 31 August 2021) database was used to analyze the association between gene expression and tumor grades in cancers. UALCAN (Ualcan.path.uab.edu/analysis, accessed on 31 August 2021) was used to analyze the protein expression in KIRC.

### 4.4. Activity Pathway Analysis Based on Blood Group Gene Expression

The GSCALite (http://bioinfo.life.hust.edu.cn/web/GSCALite/, accessed on 31 August 2021) database was used to analyze the degree of antigen gene activation or inhibition in the canonical cancer-pathways. Samples were divided into two subgroups (High and Low) according to median level of gene expression. The difference of pathway activity score (PAS) between subgroups was defined using a student *t*-test. The *p* value was adjusted using FDR. FDR ≤ 0.05 is considered as significant. When PAS A (A subgroup) > PAS B (B subgroup), we considered that gene A may have an activating effect to a pathway, otherwise it had an inhibiting effect to a pathway.

### 4.5. Gene Biological Functions of SEMA7A

To evaluate the biological functions of *SEMA7A*, differentially expressed genes (DEGs) between the two *SEMA7A* risk groups (low and high *SEMA7A* KIRC sub-groups, cut-off: 50%) were identified with limma R package, and then analyzed in the DAVID database to predict the gene ontology (GO) annotation and KEGG pathway.

### 4.6. Statistical Analysis

SPSS 22.0 and GraphPad Prism 8.0 software were used for statistical analysis. A *t*-test was conducted to explore the association between gene expression and clinicopathological factors. Kaplan–Meier curves were plotted using TCGA-KIRC data with GraphPad Prism 8.0 using the median gene expression as the subgrouping cut-off. A log-rank test was used to examine the significance of differences. Univariate and multivariate Cox regression models were used to assess the prognostic role using SPSS. Risk factors firstly selected by conducting a univariate analysis were subjected to a multivariate Cox regression analysis. A value of *p* < 0.05 was considered statistically significant.

## Figures and Tables

**Figure 1 ijms-23-08799-f001:**
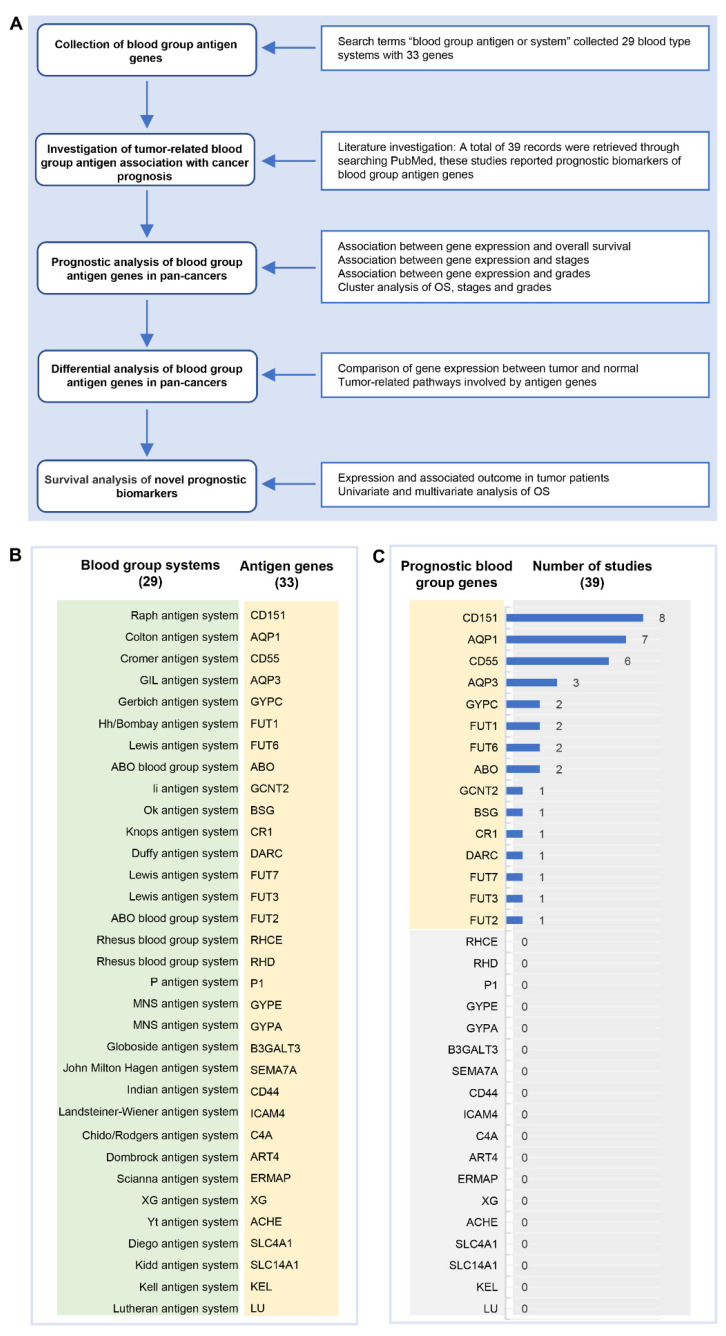
The flow diagram of this study. (**A**) The schematic diagram shows how the analysis route of the prognostic value of blood group antigen genes in pan-cancers was calculated. (**B**) The chart shows that 33 blood group antigen genes in 29 blood group systems were analyzed in this study. (**C**) The frequency of published records of blood group antigens as cancer prognostic markers.

**Figure 2 ijms-23-08799-f002:**
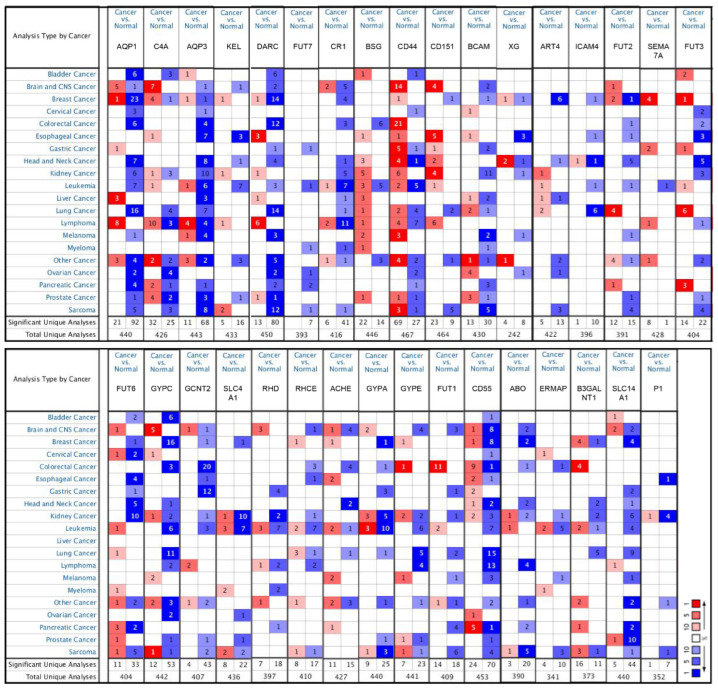
The differential expression of the blood group antigen system gene mRNA expression in pan-cancers. The threshold was set using following parameters: |log2(fold change)| ≥ 1 and *p* value < 0.05. The number in the cell represents the number of datasets that meets the thresholds. The color intensity (red or blue) is directly proportional to the significance level of upregulation or downregulation, respectively.

**Figure 3 ijms-23-08799-f003:**
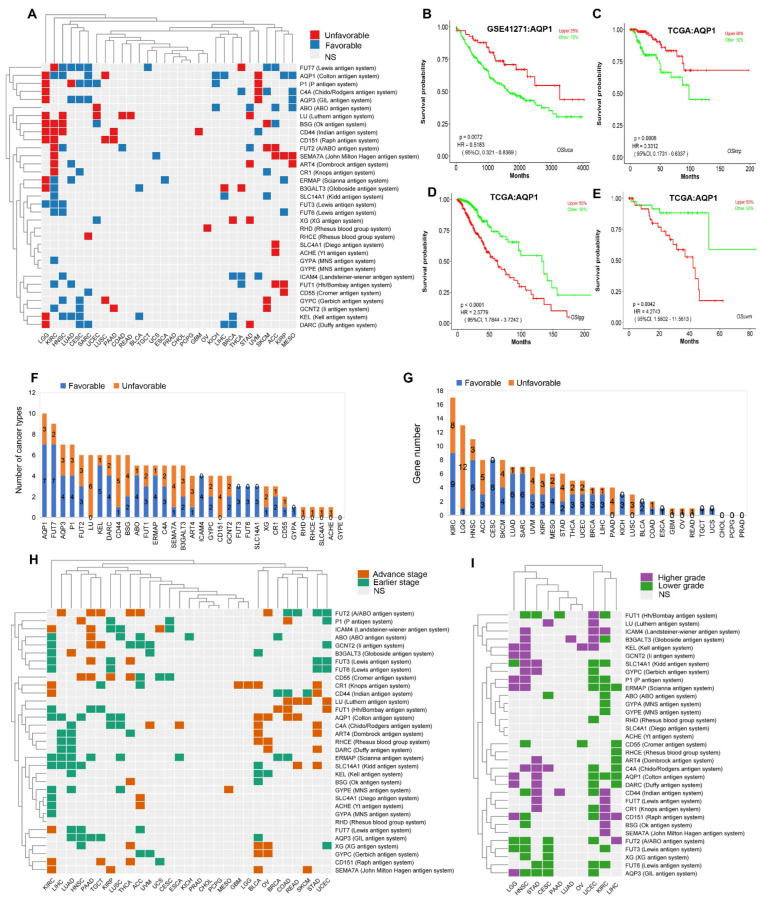
The relationship between the expression of blood group antigen system genes and prognostic clinical features in different types of cancers in TCGA dataset. Significant correlation is shown in color (*p* < 0.05). (**A**) Association between gene expression and overall survival across human cancers. Unsupervised hierarchical clustering and heatmap for the 33 blood group antigens across 30 cancers based on LOGpc results. Red represents unfavorable outcome (blue represents favorable), i.e., the gene is associated with shorter (or longer) survival. NS: Not Significant. (**B**–**E**) High *AQP1* expression was related to good OS in LUCA and KIRP, to poor OS in LGG and UVM. (**F**) The number of cancer types with prognosis potency. (**G**) The number of each blood group gene in the prognosis of different cancer types. (**H**) Association between gene expression and cancer stage across human cancers. Earlier/Advance: the gene is associated with earlier/advance stage (Spearman correlation test: *p* < 0.05). NS: Not Significant. (**I**) Association between gene expression and tumor grade across human cancers. Lower/Higher: the gene is associated with lower/higher grade (Spearman correlation test: *p* < 0.05). NS: Not Significant. Cancer types: ACC (adrenocortical carcinoma), BLCA (bladder urothelial carcinoma), BRCA (breast invasive carcinoma), CESC (cervical squamous cell carcinoma and endocervical adenocarcinoma), CHOL (cholangiocarcinoma), COAD (colon adenocarcinoma), ESCA (esophageal carcinoma), GBM (glioblastoma multiforme), HNSC (head and neck squamous cell carcinoma), KICH (kidney chromophobe), KIRC (kidney renal clear cell carcinoma), KIRP (kidney renal papillary cell carcinoma), LGG (brain lower grade glioma), LIHC (liver hepatocellular carcinoma), LUAD (lung adenocarcinoma), LUSC (lung squamous cell carcinoma), MESO (mesothelioma), OV (ovarian serous cystadenocarcinoma), PAAD (pancreatic adenocarcinoma), PCPG (pheochromocytoma and paraganglioma), PRAD (prostate adenocarcinoma), READ (rectum adenocarcinoma), SARC (sarcoma), SKCM (skin cutaneous melanoma), STAD (stomach adenocarcinoma), TGCT (testicular germ cell tumors), THCA (thyroid carcinoma), UCEC (uterine corpus endometrial carcinoma), UCS (uterine carcinosarcoma), and UVM (uveal melanoma).

**Figure 4 ijms-23-08799-f004:**
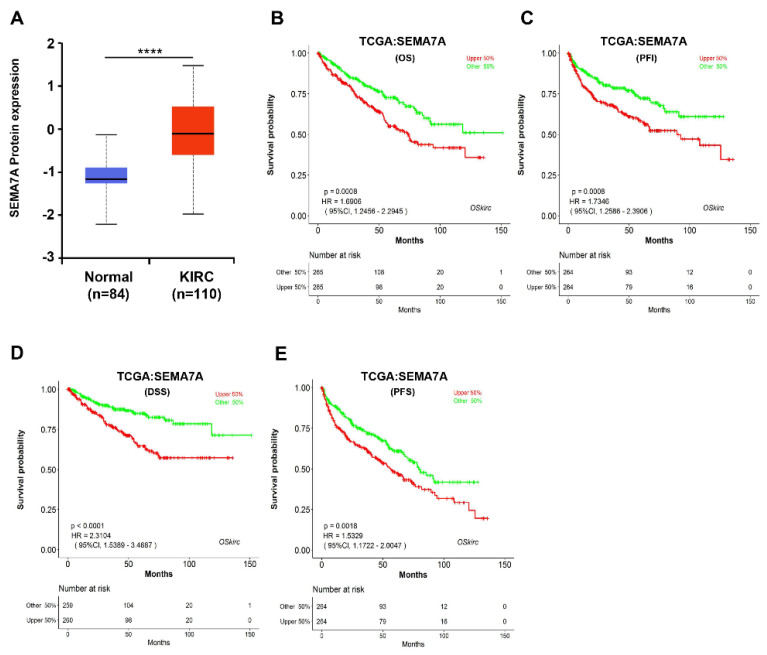
*SEMA7A* is highly expressed in KIRC cases and is associated with patients’ bad prognosis. (**A**) SEMA7A protein levels in normal and primary tumors. Log2 Spectral count ratio values from CPTAC were first normalized within each sample profile, then normalized across samples (****, *p* < 0.0001). (**B**–**E**) Kaplan–Meier curves of overall survival (OS), progression free interval (PFI), disease specific survival (DSS) and progression free survival (PFS) for KIRC patients with low and high *SEMA7A* mRNA expression. *SEMA7A* mRNA expression is associated with poor overall survival in the TCGA dataset of 523 KIRC patients (*p* < 0.05).

**Figure 5 ijms-23-08799-f005:**
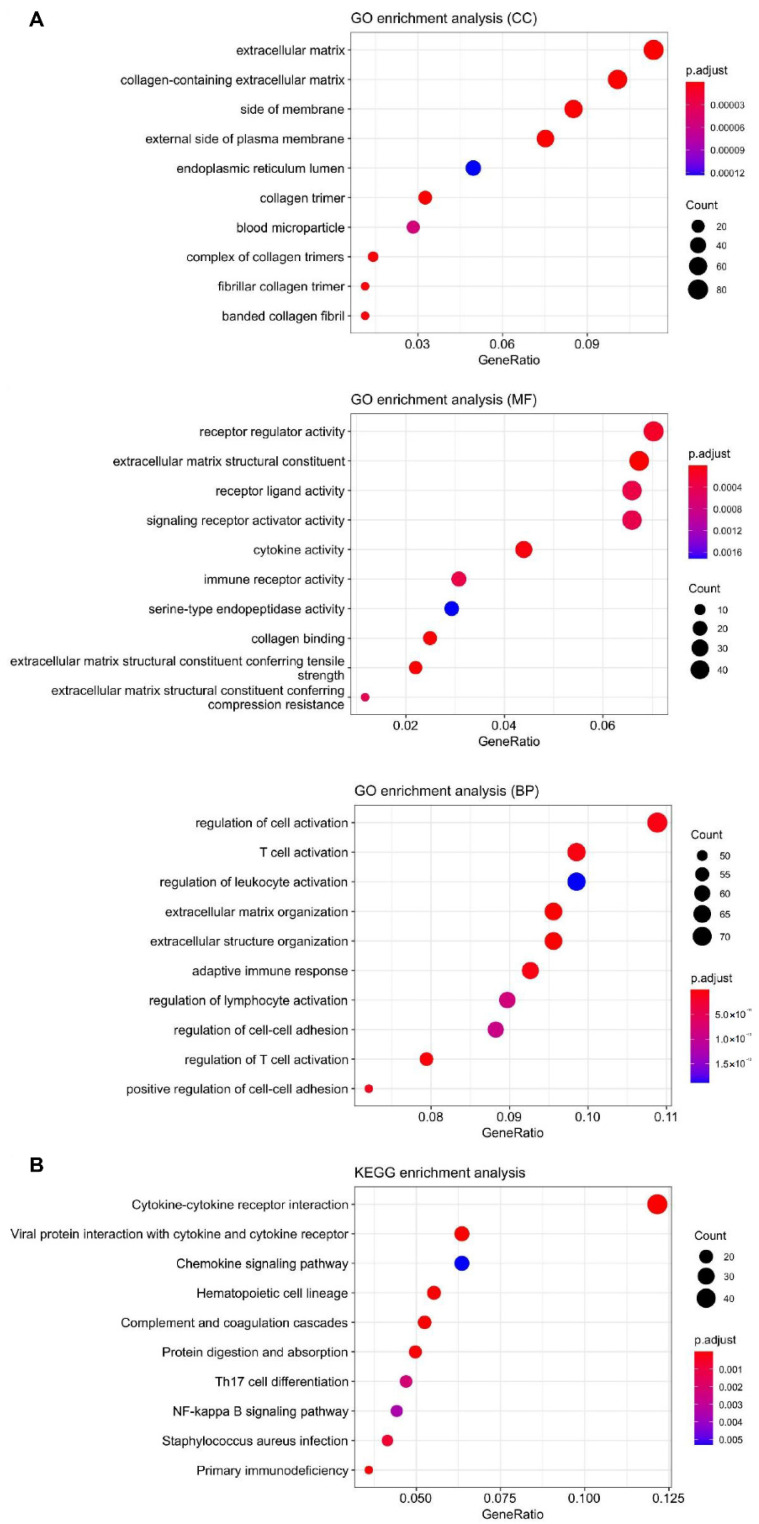
Functional and pathway enrichment analyses of DEGs between low and high *SEMA7A* KIRC subgroups (cut-off: 50%). (**A**) GO enrichment analysis in cellular components (CC), molecular functions (MF) and biological processes (BP). (**B**) KEGG enrichment analysis of DEGs.

**Table 1 ijms-23-08799-t001:** The association between *SEMA7A* expression and the demographic and clinicopathological parameters of patients with primary KIRC in the TCGA.

Parameters		Sample (N = 533)	*SEMA7A* mRNA Expression		χ^2^	*p* Value
			High (N = 266)	Low (N = 267)		
Age (Mean ± SD)		533	59.63 ± 12.13	61.62 ± 12.10		0.059 *
Gender	Male	345	185	160	5.405	0.020
	Female	188	81	107		
Grade	G1–G2	243	94	149	25.170	<0.0001
	G3–G4	282	171	111		
	Unknown	8	1	7		
TNM Stage	I–II	324	139	185	15.450	<0.0001
	III–IV	207	125	82		
	Unknown	2	2	0		
Smoking history	1	47	22	25	0.162	0.687
	2/3/4/5	40	17	23		
	Unknown	446	227			
Living status	Living	358	156	202	17.480	<0.0001
	Dead	175	110	65		

Smoking history: 1. lifelong non-smoker; 2. current smoker; 3. current reformed smoker (for >15 years); 4. Current reformed smoker (for ≤15 years); 5. current reformed smoker (duration not specified). * *t*-test.

**Table 2 ijms-23-08799-t002:** Univariate and multivariate analysis of OS in patients with primary KIRC.

Parameters	Univariate Analysis			Multivariate Analysis		
	HR	95% CI	*p* Value	HR	95% CI	*p* Value
Age (≥60 vs. <60 years)	1.813	1.324–2.484	<0.0001	1.564	1.136–2.153	0.006
Gender (Male vs. Female)	0.946	0.694–1.290	0.726	-	-	-
Grade (G3–G4 vs. G1–G2)	2.683	1.904–3.780	<0.0001	2.901	2.066–4.074	0.005
TNM Stage (III–IV vs. I–II)	3.923	2.851–5.399	<0.0001	1.682	1.169–2.422	<0.0001
Smoking (2/3/4/5 vs. 1)	0.810	0.265–2.479	0.712	-	-	-
*SEMA7A* expression (High vs. Low)	1.892	1.389–2.578	<0.0001	1.497	1.089–2.057	0.013

## Data Availability

Not applicable.

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
