# Peer review of "A Comprehensive Prognostic Analysis of Tumor-Related Blood Group Antigens in Pan-Cancers Suggests That SEMA7A as a Novel Biomarker in Kidney Renal Clear Cell Carcinoma"

_ijms, 2022, doi:10.3390/ijms23158799_

Round 1

Reviewer 1 Report

Dear authors, greetings for your big work.

I suggest to do tuo different manuscript if it is possibile. Or Can you cite SAMA7A in the title. Because your big date are on it. And because you suggest that protein as a new potential prognostic marker in KIRC.

Reviewer 2 Report

The manuscript by Y. Wang et al is about study whether the blood group antigens are associated with cancer prognosis. The authors chose 29 blood type systems with 33 genes and analyzed differential gene expression of them between tumor and normal tissue regions. Interestingly, the results show that SEMA7A may be a novel marker for poor prognosis in KIRC was shown. From the results, it seems that there are other candidates of markers. Naturally, it is necessary to investigate not only the expression level of candidate genes but also the expression amount of proteins in the future. In addition, glycosyltransferases are abundant in the genes examined and their expression is increased or decreased, so further new markers can be found by considering changes in glycan structures. I hope that the authors’ own discussion on the above will be added.

Round 2

Reviewer 1 Report

ok thank you for improve this manuscript and to have changed the title.